# Generating colorblind-friendly scatter plots for single-cell data

Tejas Guha[1], Elana J Fertig[2,3,4,5,6]*, Atul Deshpande[2,3,4]*

[1]Department of Electrical and Computer Engineering, A. James Clark School of Engineering, The University of Maryland, College Park, United States; [2]Department of Oncology, Sidney Kimmel Comprehensive Cancer Center, Johns Hopkins University School of Medicine, Baltimore, United States; [3]Convergence Institute, Johns Hopkins University, Baltimore, United States; [4]Bloomberg-Kimmel Immunotherapy Institute, Johns Hopkins University School of Medicine, Baltimore, United States; [5]Department of Applied Mathematics and Statistics, Johns Hopkins University, Baltimore, United States; [6]Department of Biomedical Engineering, Johns Hopkins University School of Medicine, Baltimore, United States

**Abstract** Reduced-dimension or spatial in situ scatter plots are widely employed in bioinformatics papers analyzing single-cell data to present phenomena or cell-conditions of interest in cell groups. When displaying these cell groups, color is frequently the only graphical cue used to differentiate them. However, as the complexity of the information presented in these visualizations increases, the usefulness of color as the only visual cue declines, especially for the sizable readership with color-vision deficiencies (CVDs). In this paper, we present scatterHatch, an R package that creates easily interpretable scatter plots by redundant coding of cell groups using colors as well as patterns. We give examples to demonstrate how the scatterHatch plots are more accessible than simple scatter plots when simulated for various types of CVDs.

## Editor's evaluation

This manuscript demonstrates a beneficial R package that provides a valuable pattern and overlay framework for producing colorblind-friendly scatter plots for the field. This work will be an extraordinary resource to the single-cell genomics community and of broad interest to many biomedical scientists, especially to viewers with color-vision deficiency. This attempt should become a new standard in the scientific community that strives to achieve greater inclusiveness.

*For correspondence:
ejfertig@jhmi.edu (EJF);
adeshpande@jhu.edu (AD)

## Introduction

Data visualization is a key component in the presentation of single-cell analyses with multiple cell groups representing factors such as cell types, states, and so forth. Color is commonly used as the only visual cue in low-dimensional scatter plots (e.g., tSNE, UMAP, etc.) or in situ spatial plots of single-cell data. We either use colormaps to represent values on a continuum or distinct colors to identify different cell groups. However, with the increasing complexity of the information being represented in these scatter plots, the ability of the readers to distinguish between the colors decreases, diminishing the interpretability of the visualizations. This problem is exacerbated for the approximately 8% of male and 0.5% of female readers who have some type of color-vision deficiency (CVD) (*Wong, 2011*). Over the course of the last decade, we have seen a number of papers (*Wong, 2011*; *Katsnelson, 2021*; *Crameri et al., 2020*; *Wong, 2010*) providing guidelines for the effective use of colors to create accessible visualizations. More recently, software packages have also been developed

that either simulate different CVDs (*Ou, 2021*), or use colorblind-friendly color palettes (*Bunis et al., 2020*; *Steenwyk and Rokas, 2021*). However, the rules for choosing accessible color palettes may change depending on the type of CVD. For example, protanopes lack the photoreceptors of red light, whereas deuteranopes lack green photoreceptors (*Deeb, 2005*). We can overcome this problem by using strategies and software solutions that reduce the dependence of visualizations on colors by 'redundant coding' (*Wong, 2011*; *Color Universal Design, 2008*; *Oliveira, 2013*) using other visual cues such as line types, point shapes, and hatched patterns over areas.

Single-cell or spatial omics data visualizations often contain scatter plots with a mixture of varying point distributions. Although simpler strategies for redundant coding already exist, they are only suitable for specific types of scatter plots. For example, we can combine colors with point shapes (*Color Universal Design, 2008*) in sparse scatter plots, but the point shapes are not apparent when the points are densely clustered together. On the other hand, distinct hatched patterns (*Wong, 2011*; *Oliveira, 2013*; *Fc and Davis, 2022*) overlaid over dense regions of the scatterplot can be used as a visual cue, but this strategy is not well suited for sparsely distributed points. However, hatched patterns can be easily added to violin plots or alluvial plots using R packages such as ggpattern to improve their accessibility.

We present *scatterHatch*, an R package for generating easily interpretable scatter plots by redundant coding of point groups using patterns and colors. scatterHatch avoids the drawbacks of the simpler strategies discussed before and easily handles point visualizations that contain mixtures of sparse and dense point distributions. Furthermore, we generate example reduced-dimension and spatial scatterHatch plots. Using the same CVD-friendly color palettes, we simulate the perception and accessibility of our scatterHatch plots against standard scatter plots for various types of CVDs.

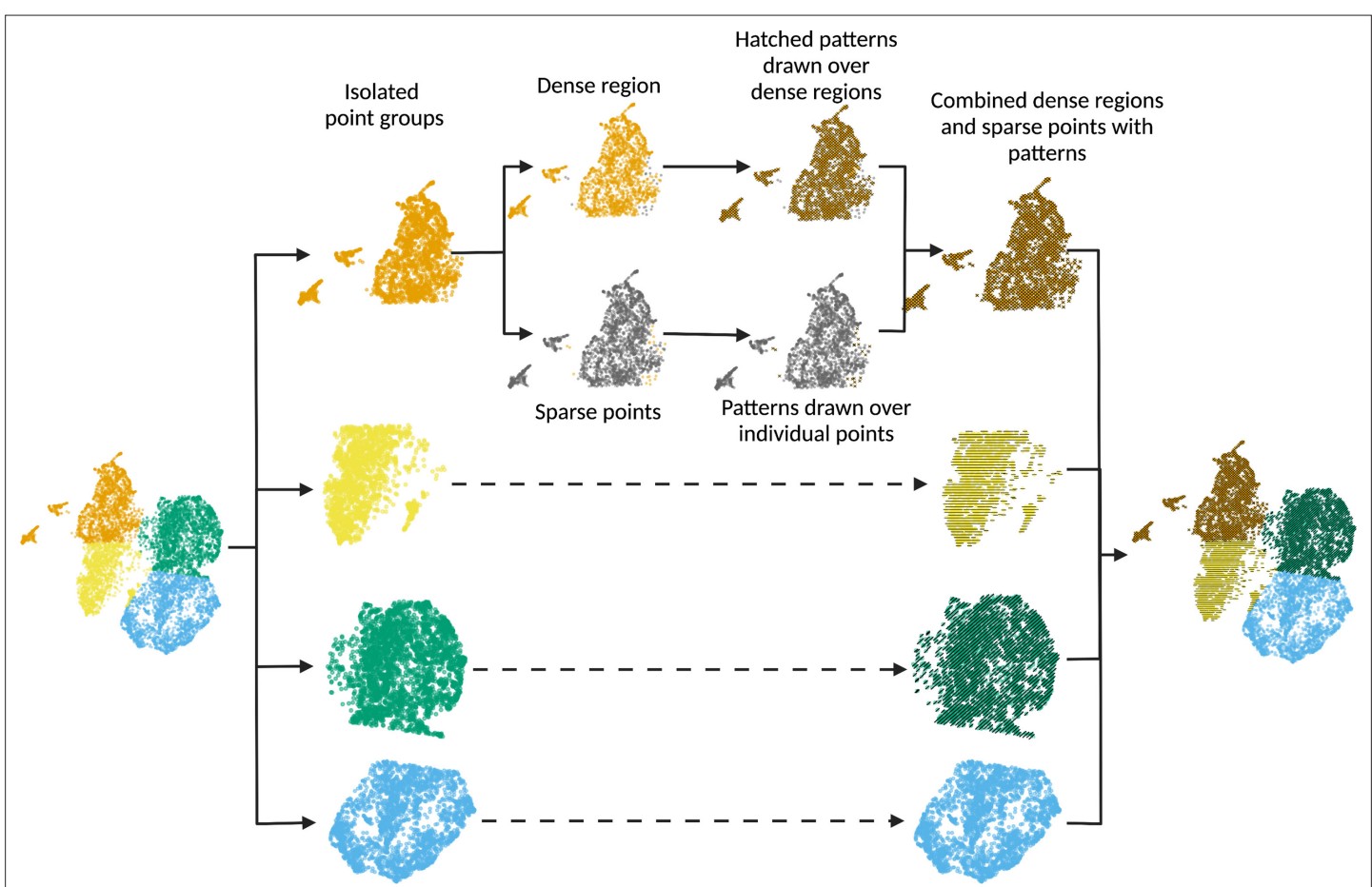

**Figure 1.** scatterHatch generates accessible scatter plots by redundant coding of point groups using colors and patterns. For every point group, scatterHatch separates sparsely distributed points from the dense clusters. scatterHatch plots coarse patterns over the dense clusters and individually plots patterns over the sparse points. Created using Biorender.com.

## Results

### scatterHatch adds patterns to scatter plots with mixtures of dense and sparse regions

In this paper, we present scatterHatch, a R/Bioconductor package to generate colorblind-friendly point visualizations commonly used in single-cell and spatial bioinformatics data analyses. scatter-Hatch greatly enhances the accessibility of low-dimensional scatter plots and in situ spatial plots of single-cell and spatial omics by using a combination of colors and patterns. A scatterHatch plot effectively represents mixtures of varying point distributions by using simple patterns which are easily plotted over dense clusters as well as sparsely distributed points.

Figure 1 shows the scatterHatch workflow. The minimum required input to scatterHatch is a data frame containing the x, y coordinates and the condition or factor to be visually represented. The output of scatterHatch is a ggplot2 object representing a scatter plot with colors and patterns assigned for each factor. Each point pertaining to a factor is classified as either belonging to a dense cluster or as an individual sparse point. scatterHatch plots coarse patterns over the dense point clusters and individually plots a matching pattern over each sparse point. Users have the option to bypass the in-built sparse point detector by providing a list of sparse points as input to scatterHatch. scatterHatch has six default patterns—horizontal, vertical, right diagonal, left diagonal, checkers and crisscross—in addition to supporting a 'blank' pattern or color-only mode. The choice of patterns is intentionally limited to those achievable by simple line segments which are suitable for both individual points or large regions of dense point clusters. The default color palette uses 40 high-contrast CVD-friendly colors imported from the *dittoSeq* package (**Bunis et al., 2020**). To advanced users, scatterHatch extends the ability to customize patterns by specifying the type (e.g., solid, dashed, dotted, etc.), color, and thickness of the lines used in the patterns. Furthermore, users can also generate new patterns composed of one or more lines by providing a list of corresponding line angles and aesthetics as input.

### Improving the accessibility of scatter plots for all types of color vision deficiencies

Here, we demonstrate the accessibility of a reduced-dimension UMAP scatter plot of single-cell data generated using scatterHatch from the perspective of different CVDs. Specifically, 10,000 cells were selected at random from single-cell data collected from a resection specimen (**Lin et al., 2018**) of Pancreatic Ductal Carcinoma (PDAC) and adjacent normal tissues. The reduced-dimension coordinates are calculated using the UMAP algorithm (**Becht et al., 2018**) and the cells are classified into four groups using K-means clustering of the UMAP coordinates. A scatterHatch plot was generated where a color and pattern were assigned to each cell group. Subsequently, we used the cvdPlot function from the R package colorblindness to simulate common CVDs such as deuteranomaly (red-green colorblindness), protanomaly (blue-yellow colorblindness), and monochromacy (complete color blindness or grayscale vision).

Figure 2 compares the accessibility of the UMAP scatter plot when compared to a scatterHatch plot as perceived by individuals with normal vision (**Figure 2A**), deuteranomaly (**Figure 2B**), protanomaly (**Figure 2C**), and monochromacy (**Figure 2D**), respectively. Each simulated visualization also includes an inset with a zoomed-in view of a region with sparse points from the distinct cell groups. As shown in the figure, the addition of the patterns makes it much easier to distinguish between factors for all types of CVD. In the zoomed-out view, we can readily distinguish between large dense clusters associated with each cell group. Similarly, the zoomed-in view demonstrates how plotting the patterns individually over sparse points enables us to distinguish them from adjacent points from other cell groups.

### Increasing the accessibility of scatter plots with large number of cell groups

The benefits of enhanced accessibility are not just limited to individuals with CVD. Different backgrounds can cause the same color to be perceived differently, or for two different colors to be perceived as the same (**Wong, 2010**). When publishing a plot with few colors, the authors can appropriately assign distinct colors to individual cell groups to avoid confusing color perceptions. As the number of colors in a scatter plot increases, however, the ability to choose distinct colors as well as to

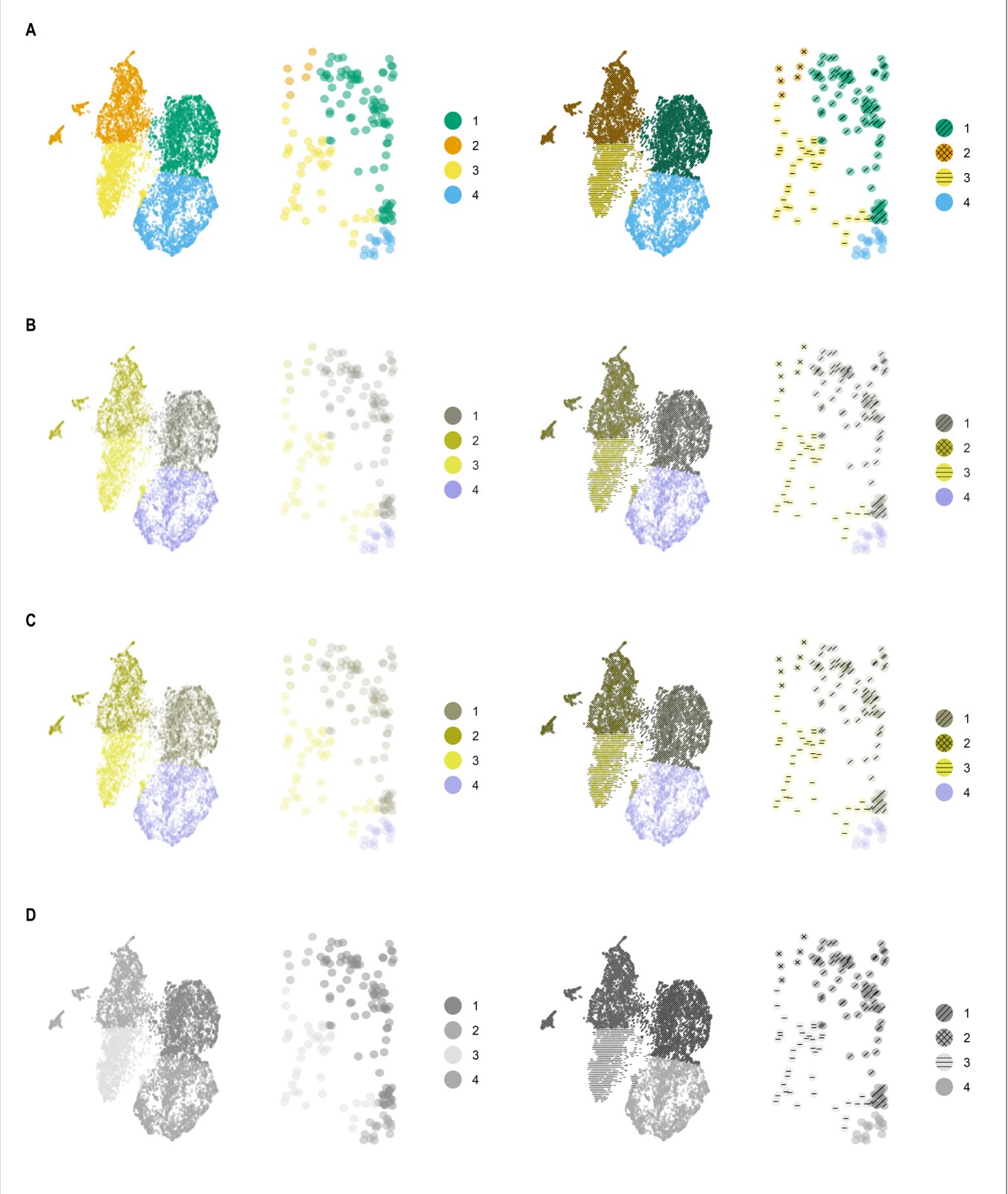

**Figure 2.** scatterHatch plots are more accessible compared to scatter plots to individuals with CVD. Simulated perception of a UMAP scatter plot compared with a scatterHatch plot by individuals with (**A**) normal color vision, (**B**) deuteranomaly, (**C**) protanomaly, and (**D**) monochromacy, with the insets showing a magnified sparse region showing patterns assigned to individual cells. Despite the change in color perception, readers have access to secondary visual information in the form of patterns to help interpret the data. CVD, color-vision deficiency.

control the relative distribution of these colors in the plot is severely hampered, leading to a higher probability of color misperception. Redundant coding with patterns facilitated the interpretation of such plots for all readers.

*Figure 3A* shows a spatial scatter plot of the cells from the PDAC resection specimen (*Lin et al., 2018*) color-coded by the frame number in the microscopy image (82 groups), with a corresponding scatterHatch plot having redundant coding using both color and pattern. For each type of plot, the groups are colored using 82 distinct colors from the "Muted Nine" colorblind-friendly color palette from the *ggpubfigs* (*Steenwyk and Rokas, 2021*) package. Using a similar approach to that used in *Figure 2*, we simulate the perceptibility of this figure for different CVDs. We see that the addition of patterns facilitates the interpretation of the visualizations.

## User-programmable aesthetics and patterns further increase the addressable dimensionality of scatterHatch

Combining the 40 colors and the 7 patterns provided in the default settings, scatterHatch is already capable of visualizing 280 patterns. Users can input custom color palettes with higher number of colors. In addition, advanced users can customize patterns by choosing line types, line colors, and line widths to achieve a broader pattern library. Finally, scatterHatch also facilitates the introduction of new patterns composed of one or more lines by providing a list of line angles and custom aesthetics. For example, in *Figure 4*, the different cell groups are represented using patterns with custom line types (PDAC cell group), custom line colors (Other and Pancreas cell groups), and completely new patterns (Small Intestine cell group). *Table 1* shows the parameters that can be used to either customize the aesthetics of a pattern or to create new patterns.

## Discussion

We present *scatterHatch,* a R/Bioconductor package for generating colorblind-friendly scatter plots of embeddings for single-cell and spatial datasets. scatterHatch enables users to generate *scatterHatch* plots—scatter plots with both a color and a pattern as visual cues. These plots are aesthetically pleasing as well as highly accessible to a broad readership including those with color vision deficiencies. scatterHatch plots are compatible with point distributions that are sparse, dense, as well as mixtures of both. We demonstrate how scatterHatch plots have better accessibility than simple scatter plots in low dimensional embeddings (e.g., PCA, UMAP, and TSNE) as well as spatial plots of cells in the tissue with up to 82-cell groups. As the number of cell groups increases, the benefits of scatterHatch plots extend even to readers with normal vision. In future work, we will make enhancements to the algorithms and plotting functions to improve the aesthetics and accessibility of scatterHatch plots with overlapping cell groups. Additionally, the software will be extended to be compatible with single-cell and spatial data formats from commonly used bioinformatics packages such as Seurat (*Butler et al., 2018*) and scanpy (*Wolf et al., 2018*).

Despite the consensus on the need for bioinformatics visualizations that are accessible across the spectrum of color perception, the progress has been slow in terms of actually affecting this change in our publications. While well intentioned, recommendations for incorporating additional steps to ensure accessible visualizations are not sufficient by themselves. For example, simulating multiple CVDs and subjectively selecting the best possible color palette for their visualizations may not come naturally to a vast majority of researchers who have normal color vision. In fact, such strategies are themselves not practical for individuals with one type of CVD who wish to ensure accessibility for other types of CVD. Software packages, such as ggpattern, dittoSeq, and scatterHatch, remove this subjectivity to a large extent by using colorblind-friendly color palettes as default, and enabling the use of visualization strategies that reduce the dependence on color. Future work could include the development of a software suite that combines the functionalities of these packages into a comprehensive software solution for creating CVD-friendly visualizations. Meanwhile, there is also a need for standards and guidelines for creating accessible visualizations, which requires support at multiple levels—from funding agencies, journals, and developers of large-scale analysis software and visualization tools (*Speir et al., 2021*). The submission review process for R or Python packages should require that the default color palettes used by the software visualizations are colorblind friendly according to well-established accessibility standards. In addition, we should develop processes for periodically

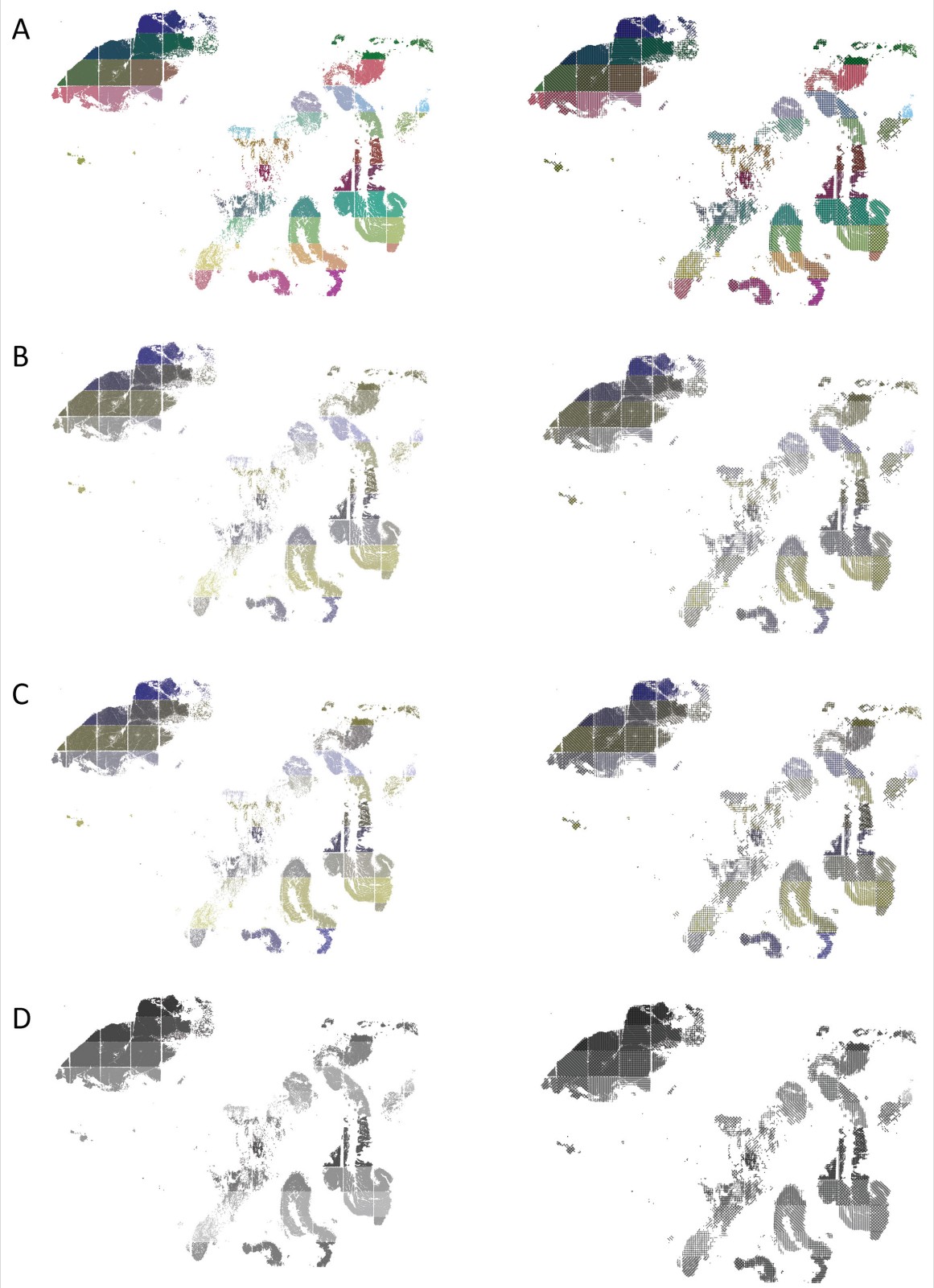

**Figure 3.** scatterHatch plots are more accessible than scatter plots for all readers when number of cell groups is high. Perception of a spatial plot of the PDAC data set with 82-cell groups compared with a corresponding scatterHatch plot as simulated for (**A**) normal color vision, (**B**) deuteranomaly, (**C**) protanomaly, and (**D**) monochromacy. As the number of colors in the scatter plot increases, its interpretability reduces even for normal color vision. The redundant coding used in scatterHatch plots results in increased accessibility. PDAC, Pancreatic Ductal Carcinoma.

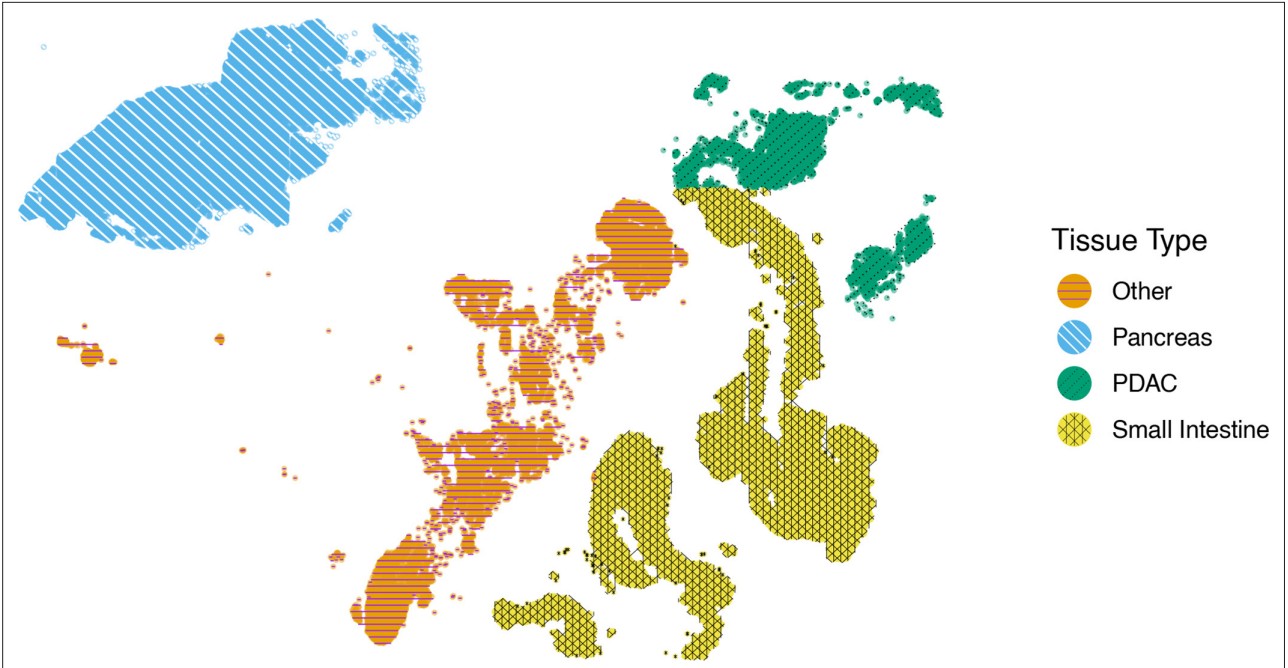

**Figure 4.** scatterHatch enables users to customize patterns. Spatial scatterHatch plot of the PDAC data set showing four tissue regions using customized patterns with a custom line type (PDAC), custom line colors (Other and Pancreas), and a completely new pattern (Small Intestine).

incorporating the best practices for accessibility introduced in new software packages into the graphical design language standards expected from newer packages. Finally, the strategies developed by scatterHatch and other recent software only address color-vision deficiencies and not other visual impairments such as double vision or complete blindness. In such cases, we need to incorporate accessibility features such as screen reader-friendly alternate texts (*Jung et al., 2022*) which describe the graphical elements of the visualizations.

## Code availability

*scatterHatch* is available on Bioconductor at https://bioconductor.org/packages/release/bioc/html/scatterHatch.html.

**Table 1.** Parameters to enable users to customize pattern aesthetics or to create new patterns.

| Name | Description | Options |
|---|---|---|
| pattern | Specifies the pattern type | Default options are 'horizontal', 'vertical', 'positiveDiagonal', 'negativeDiagonal', 'cross', 'checkers', 'blank'<br>E.g.: pattern='checkers" |
| angle | Allows users to specify line angles to be included in the pattern (enables users to create new patterns) | Numeric array with values from 0 to 180.<br>E.g.: angle=c(45, 90, 135) |
| lineWidth | Width of the lines in a pattern | Numeric - default value based on point size<br>E.g.: lineWidth=0.1 |
| lineColor | Color of the lines in a pattern | Character string specifying a color<br>E.g.: lineColor='white' |
| lineType | Type of the lines in a pattern | Character string to specify the line type from the ggplot2 package.<br>E.g.: lineType='dotted' |
| LineAlpha | Transparency of the lines in a pattern | Numeric value from 0 to 1. Default is 1.<br>Ex: lineAlpha=0.1 |

The development version is available on Github at https://github.com/FertigLab/scatterHatch, (copy archived at swh:1:rev:ae8a6b69722adb123fbcde40d12e2b2317deec2e; *Deshpande, 2022*).

The scripts used to generate the figures in the manuscript are available at https://github.com/FertigLab/scatterHatch-paper; *Deshpande, 2022*.

## Acknowledgements

This work was supported by the National Institutes of Health Grants U01CA253403, U01CA212007, and P01CA247886.

## Additional information

### Competing interests

Elana J Fertig: Reviewing editor, eLife. The other authors declare that no competing interests exist.

### Funding

| Funder | Grant reference number | Author |
| --- | --- | --- |
| National Cancer Institute | U01CA253403 | Elana J Fertig |
| National Cancer Institute | U01CA212007 | Elana J Fertig |
| National Cancer Institute | P01CA247886 | Elana J Fertig |

The funders had no role in study design, data collection and interpretation, or the decision to submit the work for publication.

### Author contributions

Tejas Guha, Conceptualization, Data curation, Software, Validation, Investigation, Visualization, Methodology, Writing – original draft, Writing – review and editing; Elana J Fertig, Resources, Supervision, Funding acquisition, Writing – original draft, Writing – review and editing; Atul Deshpande, Conceptualization, Software, Supervision, Validation, Visualization, Methodology, Writing – original draft, Writing – review and editing

### Author ORCIDs

Elana J Fertig http://orcid.org/0000-0003-3204-342X
Atul Deshpande http://orcid.org/0000-0001-5144-6924

### Decision letter and Author response

Decision letter https://doi.org/10.7554/eLife.82128.sa1
Author response https://doi.org/10.7554/eLife.82128.sa2

## Additional files

### Supplementary files
• MDAR checklist
• Supplementary file 1. 'scatterHatch user guide'.

### Data availability

The current manuscript is a computational study, so no new data have been generated for this manuscript. The scripts used for generating the figures in this manuscript are available at https://github.com/FertigLab/scatterHatch-paper, (copy archived at swh:1:rev:ae8a6b69722adb123fbcde40d12e2b2317deec2e).

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
