## [Editor Report]

This manuscript demonstrates a beneficial R package that provides a valuable pattern and overlay framework for producing colorblind-friendly scatter plots for the field. This work will be an extraordinary resource to the single-cell genomics community and of broad interest to many biomedical scientists, especially to viewers with color-vision deficiency. This attempt should become a new standard in the scientific community that strives to achieve greater inclusiveness.

---

## [Decision Letter]

**Decision letter after peer review:**

Thank you for submitting your article "scatterHatch: an R/Bioconductor package for generating colorblind-friendly scatter plots for single-cell data" for consideration by *eLife*. Your article has been reviewed by 2 peer reviewers, one of whom is a member of our Board of Reviewing Editors, and the evaluation has been overseen by Mone Zaidi as the Senior Editor. The following individual involved in the review of your submission has agreed to reveal their identity: Daniel G Bunis (Reviewer #2).

Essential revisions:

1) In Figure 1, it is a little hard to see the yellow-colored points in the sparse points demonstration. Perhaps the colors could be cycled, or one of the other regions could be used, in order for a darker color to be used for this demonstration.

2) The manuscript appears well written and where there are shortcomings would be in helping inexperienced r users navigate the add-on package. I would recommend a supplementary guide that helps novice users install and use the package--this would help strive for greater inclusiveness of individuals with varying levels of skill in r.

3) The pattern-overlay framework could be expanded and applied to other plots such as alluvial plots, violin plots, etc in addition to dim plots.

*Reviewer #2 (Recommendations for the authors):*

scatterHatch seems a well-constructed R package, and the manuscript is well-written. I believe this manuscript will be a very valuable addition to the field, especially because scatterHatch's system provides aid to viewers with monochromatic vision, an aid that even other common CVD-aware visualization tools fail to provide.

---

## [Author Response]

Essential revisions:1) In Figure 1, it is a little hard to see the yellow-colored points in the sparse points demonstration. Perhaps the colors could be cycled, or one of the other regions could be used, in order for a darker color to be used for this demonstration.

We have revised the illustration in Figure 1 to show a different cell group with darker colors for the demonstration of the scatterHatch workflow.

2) The manuscript appears well written and where there are shortcomings would be in helping inexperienced r users navigate the add-on package. I would recommend a supplementary guide that helps novice users install and use the package--this would help strive for greater inclusiveness of individuals with varying levels of skill in r.

We agree that a user guide will be really helpful! We are including a supplementary guide based on the scatterHatch package vignette in the revised submission.

3) The pattern-overlay framework could be expanded and applied to other plots such as alluvial plots, violin plots, etc in addition to dim plots.

Such pattern overlays can be obtained using existing packages like the ggpattern package available on github. However, we agree with the reviewer’s suggestion that future work could include creating a multi-purpose colorblind friendly visualization software, which include CVD-friendly strategies such as scatterHatch, ggpattern, etc.